# Total Phenolic Levels, In Vitro Antioxidant Properties, and Fatty Acid Profile of Two Microalgae, *Tetraselmis marina* Strain IMA043 and Naviculoid Diatom Strain IMA053, Isolated from the North Adriatic Sea

**DOI:** 10.3390/md20030207

**Published:** 2022-03-12

**Authors:** Riccardo Trentin, Luísa Custódio, Maria João Rodrigues, Emanuela Moschin, Katia Sciuto, José Paulo da Silva, Isabella Moro

**Affiliations:** 1Centre of Marine Sciences, Faculty of Sciences and Technology, University of Algarve, Ed. 7, Campus of Gambelas, 8005-139 Faro, Portugal; riccardo.trentin.2@studenti.unipd.it (R.T.); mary_p@sapo.pt (M.J.R.); jpsilva@ualg.pt (J.P.d.S.); 2Department of Biology, University of Padova, Via U. Bassi 58/B, 35131 Padova, Italy; emanuela.moschin@unipd.it; 3Department of Chemical, Pharmaceutical and Agricultural Sciences, University of Ferrara, Via Luigi Borsari 46, 44121 Ferrara, Italy; katia.sciuto@unife.it

**Keywords:** biodiesel, microalgal biotechnology, natural antioxidants

## Abstract

This work studied the potential biotechnological applications of a naviculoid diatom (IMA053) and a green microalga (*Tetraselmis marina* IMA043) isolated from the North Adriatic Sea. Water, methanol, and dichloromethane (DCM) extracts were prepared from microalgae biomass and evaluated for total phenolic content (TPC) and in vitro antioxidant properties. Biomass was profiled for fatty acid methyl esters (FAME) composition. The DCM extracts had the highest levels of total phenolics, with values of 40.58 and 86.14 mg GAE/g dry weight (DW in IMA053 and IMA043, respectively). The DCM extracts had a higher radical scavenging activity (RSA) than the water and methanol ones, especially those from IMA043, with RSAs of 99.65% toward 2,2′-azino-bis(3-ethylbenzothiazoline-6-sulfonic acid)diammonium salt (ABTS) at 10 mg/mL, and of 103.43% against 2,2-diphenyl-1-picrylhydrazyl (DPPH) at 5 mg/mL. The DCM extract of IMA053 displayed relevant copper chelating properties (67.48% at 10 mg/mL), while the highest iron chelating activity was observed in the water extract of the same species (92.05% at 10 mg/mL). Both strains presented a high proportion of saturated (SFA) and monounsaturated (MUFA) fatty acids. The results suggested that these microalgae could be further explored as sources of natural antioxidants for the pharmaceutical and food industry and as feedstock for biofuel production.

## 1. Introduction

The large amount of resources provided by the marine environment constitute the basis of many economic activities and, according to the Green Paper—Towards a Future Maritime Policy for the Union, the exploitation of the marine biodiversity will contribute to many industrial sectors, including those related to aquaculture, healthcare, cosmetics, food/feed, and energy [1].

In particular, algal extracts have been gaining increasing interest due to their unique chemical composition and possibility of wide industrial applications over the last few years [2,3,4]. Recent developments in genetic engineering, coupled with the enormous biodiversity of microalgae, make this group of organisms one of the most promising sources for new products and applications [4]. With the development of new culture and screening techniques, microalgal biotechnology can already meet the high demands of both the food and pharmaceutical industries [5] and is considered fundamental for the development of sustainable processes that contribute to the global bioeconomy [6]. However, marine microalgae remain largely unexplored and still represent an almost untapped reservoir of novel products and metabolites, such as lipids, with possible applications in different sectors, including aquaculture, human health, nutrition, and biofuel production [1,4,7].

Green algae (Chlorophyta) and diatoms (Bacillariophyta) are acknowledged as important sources of high-added-value natural products. However, a reliable molecular identification is critical for the bioprospection of these organisms [8]. Mistakes in sample identification can have direct implications for the interpretation of biochemical and physiological results, undermining our understanding of the biology of these species, as well as the possibility of a commercial exploitation [9]. Phylogenetically related species to *Tetraselmis marina* (IMA043) and a naviculoid diatom (IMA053) have important commercial applications. For example, *Tetraselmis* (Stein) species (green marine microalgae) are commonly used in aquaculture, due to their high nutritional value in terms of proteins and fatty acids (FA) [10]. Specifically, different strains of the species *T. marina* (Cienkowski) R.E.Norris, Hori, and Chihara are used as sources of carotenoids and FA for feed, food, and biofuel production (strain CTM 20,015) [10] and in the bioremediation of water contaminated with copper, iron, and manganese (strain AC16-MESO) [11]. Furthermore, Naviculales (Bessey) is an order of benthic diatoms comprising more than 5600 species having a similar boat-shape, and produces several bioactive compounds of commercial interest [12,13]. For example, phytosterols from *Navicula incerta* Grunow strain KMMCC B-001 induced apoptosis in hepatocarcinoma (HepG2) cells by upregulating the expression of pro-apoptotic gene (Bax, p53) while downregulating the anti-apoptotic gene Bcl-2 [14]. Other members of this order, including *Navicula* sp. strain JPCC DA0580 [15] and *Fistulifera solaris* strain JPCC DA0580, are considered a promising source of lipids for biofuel production [16].

The main goal of this work was to evaluate the potential of two microalgae strains, isolated from the North Adriatic Sea and identified combining morphological and DNA sequence data, as new sources of natural antioxidants, FA, and phenolic compounds. For this purpose, the naviculoid diatom strain IMA053 and *T. marina* strain IMA043 were isolated from water samples collected in the North Adriatic Sea (Chioggia, Italy) cultivated under laboratory conditions and identified by light microscopy and genetic analyses (amplification of the 18S rDNA gene). The obtained dried biomass was characterized in terms of FA methyl ester contents (FAME) by gas chromatography–mass spectrometry (GC–MS), and used for the preparation of methanol, water, and dichloromethane (DCM) extracts. The extracts were evaluated for total content of phenolics (TPC) by a spectrophotometric method, and for in vitro antioxidant activity by radical-based methods (scavenging on the 1,1-diphenyl-2-picrylhydrazyl (DPPH) and 2,2′-azino-bis 3-ethylbenzothiazoline-6-sulfonic acid (ABTS) radicals) and metal-based methods (metal chelation of copper and iron, and ferric reducing activity).

With this study, we have contributed to the knowledge about the production capability of different potentially useful compounds by the genus *Tetraselmis* (phylum Chlorophyta) and the family Stauroneidaceae (phylum Bacillariophyta). In particular, besides adding data to the little information so far available for the TPC of the naviculoid family Stauroneidaceae, this was the first time that the TPC of the species *T. marina* was analyzed. In addition, the antioxidant properties of different extracts from the two investigated microalgal strains were considered by applying five complementary methods, including radical and metal-related assays. In some cases, the obtained results confirmed what was previously found for other species of the genus *Tetraselmis* and for other naviculoid species, while, in other cases, our findings were not in agreement with the evidence already reported for other phylogenetically related taxa. The same could be said for the FAMEs profiles that were obtained for both *Tetraselmis marina* (IMA043) and the naviculoid diatom (IMA053). Possible explanations for the observed discrepancies with previous studies could be explained by the choice of the growing conditions, by the polarity of the extraction solvents used in this study, and by interspecific variations. Moreover, as our results regard new and unexplored taxa, the possession of unique properties by different microalgal taxonomic entities is emphasized.

## 2. Results and Discussion

### 2.1. Molecular Analyses

To establish the taxonomical position of the strains IMA043 and IMA053, isolated from the seawater samples collected from the North Adriatic Sea, a molecular analysis was carried out, through the 18S rRNA gene. Regarding the IMA043 strain, the comparison of its 18S rRNA gene sequence with other ones available in GenBank showed the highest percentage identities with sequences of *T. marina* CCMP898 (99.90%) and *Tetraselmis* sp. KMMCC 1293 (100%). This was evident also in the ML tree generated by IQTREE, where strain IMA043 was part of the *T. marina* clade (Figure 1), confirming the identification based on morphological observations.

The comparison of the 18S rRNA sequence of strain IMA053 with the other sequences from GenBank showed the highest percentage identities with sequences of Bacillariophyta genera (*Sternimirus*, *Prestauroneis* and *Stauroneis*), all belonging to Stauroneidaceae family. In particular, the highest percentage identity (99.65%) was with a sequence of ‘Bacillariophyta sp. MBIC10102′, a strain not identified at the species or genus level. Maximum-likelihood (ML) phylogeny of 18S rRNA gene sequences of Naviculales confirmed our identification of strain IMA053 as a species belonging to the Stauroneidaceae family (Figure 2), but to better identify this strain at the genus or species level, it will be necessary to carry out further molecular analyses by using different markers (*rbc*L, *psb*C, and ITS).

### 2.2. Light Microscopy

Light microscope observations of the isolated *T. marina* IMA043 showed slightly compressed oval cells, about 12 µm long, with the posterior part wider than the apical one. Cells were shown in the anterior end of a depression, where four equal-length flagella were located, emerging from the cell in two pairs (Figure 3a). Cells were green in color with a single large cup-shaped chloroplast, sometimes located in the posterior end of the cell. A single pyrenoid was present within the chloroplast. The pyrenoid occupied a central position inside the chloroplast (Figure 3a). Moreover, in the anterior part of the cell, there was an orange-red eyespot (Figure 3a). *Tetraselmis marina* cells divided in a nonmotile vegetative stage (Figure 3b).

Naviculoid diatom IMA053 cells showed a simple shape and a bilateral symmetry. Valves were lanceolate with rounded apices in the valve view (Figure 3c). In the valve view, the central raphe system was evident (Figure 3c). Even if the valves generally offer more features useful for the identification of the species, during our observation, there were many lipid inclusions preventing the valve analysis (Figure 3d).

### 2.3. Total Phenolics Content (TPC) of the Extracts

Polyphenolic compounds are the main contributors to the antioxidant defense mechanisms of plants and algae and exhibit several interesting biological activities relevant for commercial exploitation, related to their radical scavenging properties, such as anti-inflammatory, anti-atherosclerotic, and anti-carcinogenic [17,18,19]. Therefore, this work explored the TPC of extracts from the two strains, *T. marina* IMA043 and naviculoid diatom IMA053 (Table 1). The TPC of the different samples varied from 17.80 mg GAE/g dry weight (DW) in the water extract of naviculoid diatom IMA053 to 86.14 GAE/g (DW) in the DCM extract of *T. marina* IMA043. The extraction of different phenolics relies on their polar properties and, especially, polyphenols are often more soluble in organic solvents less polar than water, for example, DCM [19,20,21].

To our best knowledge, there is no published information on the total levels of phenolics of *T. marina* and little is known regarding the phenolic content of species of the Stauroneidaceae family. This work highlighted that the TPCs of both *T. marina* IMA43 and naviculoid diatom IMA053 are higher than those described in the literature for related species, obtained with the same methodology (F–C method). In particular, methanol and hexane extracts from *T. chuii* [22], ethanol/water extracts from *T. suecica* (Kylin) Butcher [23], methanol extract from *Navicula* sp. [24], and 50% ethanol extracts from *Stauroneis* sp. LACW24 [25] showed lower levels of phenolics compared to IMA043 and IMA053. These differences could be explained by the use of solvents with different polarities for the extractions, such as hexane and chloroform, or by choosing different growing conditions that can affect the final composition of microalgae [20,21,22,23]. In fact, several environmental conditions could be stimulants for the accumulation of different phenolics, including nitrogen and phosphorus limitation, and salinity, on the microalgae culture medium [10,24]. This is because phenolic compounds act as defense against a wide range of stresses and are thus accumulated in response to these environmental conditions [25]. Moreover, variations in the TPC of extracts obtained by extraction with different solvents are ascribed to different polarities of the compounds present in the biomass. Phenols include hydroxyl groups (polar part) attached to an aromatic ring (nonpolar part) [26]. This stereochemistry distinguishes phenols according to their polarity, which has been used for the recovery of phenols from natural sources by extraction with different solvents. Therefore, the extraction yield in terms of polyphenolic components strongly depends on the nature of the solvent and the extraction is affected by the diffusion coefficient and the dissolution rate of compounds until they reach the equilibrium concentration inside the solvent [27]. Phenolic compounds usually have a more polar nature, as reported by different authors. For example, Hajimahmoodi et al. [28] and Goiris et al. [22] obtained the highest phenolic content in the most polar extract (hot water extracts) from different microalgae species, such as *Chlorella* and *Tetraselmis* sp. However, Li et al. [18] found the highest content of phenolic substances in apolar extracts, namely hexane fractions from *Chlorella* and *Nostoc* species. The recovery of certain phenolic terpenes has been referred to occur preferably with less polar solvents, while more polar solvents have been reported to extract flavonoid glycosides and higher-molecular-weight phenols [29]. In our work, the results suggest that the species being studied are rich in polyphenolic compounds of a more nonpolar nature extractable with solvents such as DCM.

### 2.4. In Vitro Antioxidant Properties

The accumulation of reactive oxygen species (ROS) generates oxidative stress, causing, for example, photosynthesis failure [30]. To cope with oxidative stress, microalgae produce different classes of antioxidant compounds (e.g., phenols, sulphated polysaccharides, carotenoids) that could be exploited as food supplements, food preservatives, nutraceuticals, or pharmaceuticals [19,22,29,30,31,32]. In this work, the extracts from *T. marina* (IMA043) and the naviculoid diatom (IMA053) were evaluated for antioxidant activity by five complementary methods, including radical and metal (copper and iron)-related assays. The less polar (DCM) extracts had the highest capability to scavenge the DPPH radical, 178.75% (*T. marina*) and 91.38% (naviculoid diatom), when compared to the methanol and water extracts (Table 2). The high percentage of DPPH scavenging activity of the DCM extracts from *T. marina* is consistent with other findings on *T. suecica* [33] and *T. chui* [34]. The water and methanol extracts from *T. marina* had no capacity to scavenge DPPH, even though other authors detected a high RSA in water extract from *Tetraselmis* sp. and in methanol extracts from *T. tetrathele* (West) Butcher and *T. chuii* Butcher [21,35,36]. However, interspecific variations and different growing conditions, such as nutrient availability and light intensity, can account for the differences observed for different species belonging to the *Tetraselmis* genus [22,33,36]. In turn, the methanol and water extracts from *T. marina* IMA043 had no capacity to scavenge the ABTS radical; however, the DCM extract showed a higher RSA toward the ABTS radical when compared with all the extracts from the naviculoid diatom (IMA053), at all tested concentrations. These results are consistent with those found in the literature, as a relatively high ABTS scavenging activity is reported in ethanol/water extracts from algae of the genus *Tetraselmis*, such as *Tetraselmis* sp. and *T. suecica* [22], and diatoms, including *Chaetoceros calcitrans* (Paulsen) H.Takano and *Phaeodactylum tricornutum* Bholin [22,37]. Moreover, studies on the antioxidant properties of other phylogenetically related diatoms, such as *Navicula incerta* [38] and *Stauroneis* sp. LACW24 [23], reported a high RSA toward DPPH and ABTS, respectively, which is in accordance with our results on the naviculoid diatom (IMA053). At the concentration of 10 mg/mL, the methanol and DCM extracts presented a moderate RSA toward the ABTS radical (35.54% and 43.83%, respectively), while the water extract had a lower antioxidant capacity (23.51%).

The FRAP of the extracts at the concentration of 10 mg/mL ranged from 21.47% and 26.75% in the water extracts from both *T. marina* (IMA043) and the naviculoid diatom (IMA053) to 150.10% in the DCM extract from the naviculoid diatom (IMA053) and 103.17% in the methanol extract of *T. marina* (Table 3). Other microalgae exhibited a strong ferrous reducing power, as reported for acetone extracts of *Navicula* sp. [39] and hydroethanolic extracts from *Tetraselmis* sp. [22]. Molecules capable of reducing iron are electron donors that reduce the oxidized intermediates of lipid peroxidation processes, acting as primary and secondary antioxidants [40]. Our results indicate that both microalgal strains contain molecules that act as electron donors stabilizing radical species and counteracting their harmful effect [41], thus preventing damage to DNA, lipids, proteins, and other biomolecules [42].

Transition metal ions, such as copper (Cu^2+^) and iron (Fe^2+^), can lead to the formation of free radicals that may cause modifications to DNA bases, enhance lipid peroxidation, and affect calcium and sulphydryl homeostasis [43]. Therefore, the use of natural products with the capacity to chelate those metals is considered a useful strategy to manage disorders related to the accumulation of such ions [44]. According to our findings, the water and DCM extracts of the naviculoid diatom (IMA053) showed moderate to high metal chelating activity, while the methanol extract had no metal chelating properties, unlike the results reported for *Navicula incerta* [38] (Table 4). However, these differences can be explained by different causes, including interspecific variations and extrinsic factors, such as light intensity, salinity, and nutrient levels, or by the different tested extracts [45]. The water extract from *T. marina* showed a high capacity to chelate Fe^2+^; a similar metal chelating activity is reported in the literature for the genus *Tetraselmis* [36].

### 2.5. Relation between Phenolic Contents of the Extracts and the In Vitro Antioxidant Properties

As phenolic compounds are considered one of the most important classes of natural antioxidants, a relation between the TPC of the two strains of Adriatic microalgae and their RSA was expected. In this sense, the linear association between TPC and the antioxidant activity obtained with the different assays (FRAP, DPPH, and ABTS) was expressed as the correlation coefficient *R*^2^. The correlation coefficient between FRAP and the TPC of the studied samples was *R*^2^ = 0.75, whereas, for the correlation between antioxidant activity (DPPH and ABTS) and TPC, it was higher (*R*^2^ = 0.85 for DPPH and *R*^2^ = 0.95 for ABTS) (Figure 4). These positive correlations suggest that 75% of the capacity of the naviculoid diatom (IMA053) and *T. marina* (IMA043) strains to reduce iron was due to the contribution of phenolic compounds, while from 85% to 95% of the antioxidant capability of these microalgae was related to the presence of phenols (Figure 4). The remaining 25% (for FRAP), 15% (for DPPH), and 5% (for ABTS) of the antioxidant activity that is not explained by the presence of phenols could be ascribed to other molecules present in the extracts with antioxidant properties [46]. Overall, our results suggest that phenolic compounds were a major contributor to the antioxidant capacities of the tested extracts.

### 2.6. FAMEs Profile

*Tetraselmis marina* (IMA043) and the naviculoid diatom (IMA053) were analyzed for FAMEs profile, by GC–MS, after direct transesterification of the dried biomass, and the results are summarized on Table 5. Biomass from both species had a prevalence of MUFA (49.81% for IMA053 and 45.41% for IMA043), followed by SFA (45.70% for IMA053 and 38.28% for IMA043) and PUFA (4.57% for IMA053 and 16.31% for IMA043). A similar FAME profile was observed for *Tetraselmis* sp. [47], while higher concentrations of PUFAs were found in *Tetraselmis* sp. IMP3 and *Tetraselmis* sp. CTP4 [48]. The FAMEs profile of IMA053 is similar to that of *Stauroneis* sp. LACW24 [49]; however, differences in PUFAs content were observed in *Navicula* sp. maintained in the MAScIR’s microalgae collection [47]. These variations in the relative proportions of the FAMEs composition of microalgae has a high phenotypic plasticity and can occur in response to different culture conditions [49,50]. In particular, an increase in PUFA levels at low culture temperatures is a common trend reported for microalgae [51]. The FA profile of *T. marina* consisted predominantly of palmitic (C16:0) and oleic (C18:1n9c) acids, in accordance with other findings for the same species [10] and for other species of the same genus, such as *Tetraselmis* sp. [47]. The main FA identified in the naviculoid diatom (IMA053) were palmitoleic acid (C16:1) and palmitic acid (C16:0), as previously reported for other closely related species, including *Navicula* UMACC 231 [52] and *Stauroneis* sp. LACW24 [49]. Among these abundant FA detected in IMA043 and IMA053, palmitic acid (C16:0) from microalgae is widely recognized as an antimicrobial agent [53,54], while palmitoleic acid (C16:1) accumulation is reported to be a remarkable characteristic for the production of biodiesel [55], and preliminary clinical trials suggested that this compound could promote weight loss, reduce cholesterol levels, and manage inflammation [56]. Oleic acid (C18:1n9c), abundant in *T. marina* IMA043, is the most common MUFA in human nutrition and emerging studies have shown that diets enriched in oleic acids can contribute to the management and prevention of obesity, and one of its derivatives (oleoylethanolamide) has been demonstrated to reduce hunger and subsequent food consumption [57].

The lipid content is an important prerequisite determining the suitability of microalgae for commercial biofuel production [58] and the high proportion of SFAs and MUFAs in the naviculoid diatom (IMA053) and *T. marina* (IMA043), suggesting their potential use for that purpose [59,60]. Other closely related species such as *N. pelliculosa* (Kützing) Hilse and *T. suecica* have already proved to be desirable for biodiesel production as they can accumulate a large mass of oils per unit volume of the microalgal broth per day [61,62]. High SFAs and MUFAs percentages, together with the absence of C18:3 in the lipid profile of the naviculoid diatom (IMA053) and *T. marina* (IMA043), ensure that both species meet European standard specifications (SFA and MUFA ≥ 12%, C18:3 ≤ 1) for biodiesel [63,64]. Particularly, the naviculoid diatom (IMA053) showed the highest oleic acid content, which is an important characteristic to produce good-quality biodiesel. It is reported that oils with high oleic acid content have a reasonable balance of fuel, including its ignition quality, combustion heat, cold filter plugging point (CFPP), oxidative stability, viscosity, and lubricity, which are determined by the structure of its component fatty esters [65].

## 3. Materials and Methods

### 3.1. Chemicals

The compounds DPPH, ABTS, and FAME standards (Supelco^®^ 37 Component FAME Mix) were purchased from Sigma (Steinheim am Albuch, Germany). Merck (Darmstadt, Germany) supplied the Folin–Ciocalteau (F–C) phenol reagent and all solvents used for chemical analyses. Additional reagents and solvents were obtained from VWR International (Leuven, Belgium).

### 3.2. Sampling, Strain Isolation and Culture Set Up

Seawater samples were collected in the North Adriatic Sea, near Chioggia, through a plankton net with a nylon mesh of 45 µm. The single-cell isolation of *Tetraselmis marina* (IMA043) and the naviculoid diatom (IMA053) was performed with a heated flame, extended, and broken Pasteur glass pipette. After isolation, cells were maintained in liquid cultures with F/2 medium [66] in a growth chamber at 16 °C, with a photosynthetic photon flux density of 35 µmol photons m^−2^ s^−1^ under a 12-h:12-h light/dark cycle. Microalgal biomass was harvested by centrifugation and immediately frozen at −20 °C. Freeze-dried biomass was obtained upon lyophilization for 24 h and stored at room temperature (RT) in the dark.

### 3.3. Genetic Analyses

Total genomic DNA of IMA043 and IMA053 was extracted from microalgal pellets, using the Genomic DNA purification kit (Fermentas, Burlington, ON, Canada) according to the manufacturer’s protocol. Amplification of the 18S rDNA gene was carried out with the primers Euk528F and EukB [67,68] under the following PCR conditions: initial denaturation at 95 °C for 5 min, followed by 30 cycles (95 °C for 50 s, 52 °C for 50 s, and 72 °C for 1 min 30 s) and final elongation at 72 °C for 8 min. The obtained PCR products were visualized with GelRed (Biotium) staining after electrophoresis in a 1% agarose gel, purified with the ExoSAP-ITTM kit (Amersham Biosciences, Piscataway, New York, NY, USA), and directly sequenced with the same primers used in the amplification reaction. Sequencing was performed at the BMR-Genomics Sequencing Service (Padua University) on both strands to ensure accuracy of the results. The final consensus sequences were assembled using the SeqMan II program from the Lasergene software package (DNAStar, Madison, WI, USA) and analyzed by similarity search using the BLAST program [69], available at the NCBI web server (www.ncbi.nlm.nih.gov/blast, accessed on 23 December 2021). The 18S rRNA sequences generated in this study were deposited in GenBank with the following accession numbers: OM319687 for naviculoid diatom IMA053 and OM319688 for *Tetraselmis marina* IMA043.

For the phylogenetic analysis, two separate datasets were created for the 18S rRNA partial sequences of the strains IMA043 and IMA053, including the new sequences generated in this study and other related sequences obtained from the NCBI archives. Each dataset was aligned with the online version of MAFFT (https://mafft.cbrc.jp/alignment/server, accessed on 27 December 2021). Maximum likelihood (ML) phylogenetic trees were computed for both datasets using the IQ-TREE webserver (http://iqtree.cibiv.univie.ac.at/, accessed on 27 December 2021) [70] using the best-fitting models (TIM3 + F + I + G4 for the IMA043-aligned dataset and GTR + F + I + G4 for the IMA053-aligned dataset) selected by the Model Finder algorithm [71] implemented in IQ-TREE. The statistical support for tree topologies was computed by performing 10,000 ultrafast bootstrap replicates [72] and the SH-aLRT branch test [73]. Trees were visualized in iTOL v5 (https://itol.embl.de, accessed on 27 December 2021) [74].

### 3.4. Light Microscopy

Cultures of the strains IMA043 and IMA053 were observed with a light microscope (LM) Leitz Dialux 22 (Wetzlar, Germany), equipped with a digital image acquisition system.

### 3.5. Preparation of the Extracts

The extracts were prepared by an ultrasound-assisted sequential method, as follows. One gram of dried biomass was mixed with water (200 mL) and the algal cell walls were disrupted in an ultrasonic bath (USC-TH, VWR, Portugal; capacity of 5.4 L, frequency of 45 kHz, a supply of 230 V, tub heater of 400 W, temperature control made by a LED display) for 15 min. Samples were filtered (Whatman no. 4) and the residue was then sequentially extracted with methanol and DCM, as described previously. The extracts were dried under a reduced vacuum pressure at 40 °C, weighed, dissolved in the corresponding solvent (water, methanol, and DCM) at the concentration of 50 mg/mL, and stored (4 °C for the methanol and DCM extracts, −20 °C for the water extract).

### 3.6. Total Phenolics Content (TPC) of the Extracts

TPC was determined by the F–C method [75] adapted to 96-well plates according to Custódio et al. [21]. Extracts (5 µL at the concentration of 10 mg/mL) were mixed with 100 µL of 10-fold-diluted F–C reagent in distilled water. After 5 min of incubation at RT, 100 µL of sodium carbonate (75 g/L, *w*/*v*, in water) was added and incubated for 90 min at RT. The absorbance was measured at 725 nm on a microplate reader (Biochrom™ EZ Read 400, Biochrom Ltd., Cambridge, UK) and the results were expressed as gallic acid equivalents in milligrams per gram of extract (mg GAE/g dry weight, DW). A calibration curve was built using gallic acid standard solutions.

### 3.7. In Vitro Antioxidant Properties

The in vitro antioxidant properties of the extracts were evaluated by five complementary methods, including radical scavenging activity (RSA) on the free radicals (DPPH and ABTS), metal chelation of iron and copper ions, and ferric reducing power (FRAP). The extracts were tested at three different concentrations (1, 5, and 10 mg/mL). The absorbances were measured using a microplate reader (Biochrom™ EZ Read 400, Biochrom Ltd., Cambridge, United Kingdom) and the RSA and metal chelating activities were calculated as the percentage of inhibition relative to a blank-containing solvent instead of the extracts. FRAP results were expressed as inhibition (%) relative to the positive control (butylated hydroxytoluene: BHT), tested at 1 mg/mL.

#### 3.7.1. Radical-Based Assays: RSA on DPPH and ABTS

The RSA against DPPH was evaluated by the method of Brand Williams et al. [76], adapted to 96-well microplates as reported by Custódio et al. [77]. Briefly, 22 µL of the extracts was mixed with 200 µL of an ethanol DPPH solution (120 µM) in 96-well microplates. After 30 min of incubation at RT in the dark, the absorbance was measured at 515 nm. The RSA on the ABTS radical was determined according to the method described by Re et al. [78]. A stock solution of ABTS (7.4 mM) was prepared in potassium persulfate (2.6 mM), left in the dark for 12–16 h at RT, and diluted with ethanol until it reached an absorbance of 0.7 at 734 nm. For the assay, 10 µL of the extracts was mixed with 190 µL of ABTS in 96-well microplates and incubated in darkness at RT for 6 min. The absorbance was read at 734 nm. In both assays, BHT was used as the positive control at the concentration of 1 mg/mL.

#### 3.7.2. Metal-Based Assays: Ferric Reducing Activity Power (FRAP) and Metal Chelation of Iron and Copper

FRAP was evaluated according to Megías et al. [79]. In brief, 50 µL of the samples was mixed in 96-well microplates with 50 µL of potassium ferricyanide (1% in water) and 50 µL of distilled water. After 20 min of incubation in the dark at 50 °C, 50 µL of 10% TCA (trichloroacetic acid in water) and 10 µL of ferric chloride solution (0.1% in water) were added. The absorbance was measured at 700 nm after 10 min of incubation at RT, and BHT was used as the standard.

Copper chelating activity (CCA) was assessed according to Megías et al. [79]. Samples (30 µL) were mixed in 96-well microplates with 200 µL of 50 mM sodium acetate buffer (pH 6), 6 µL of pyrocatechol violet (PV, 4 mM in the acetate buffer), and 100 µL of copper sulphate (50 μg/mL in water). The absorbance was measured at 632 nm. Iron chelating activity (ICA) was determined as described by Megías et al. [79] by measuring the formation of the Fe^2+^ ferrozine complex. Briefly, 30 μL of the extracts was mixed in 96-well microplates with 200 µL of distilled water and 30 µL of an iron (II) chloride solution (0.1 mg/mL in water) and incubated for 30 min at RT. Afterward, 12.5 µL of ferrozine solution (40 mM in water) was added and the change in color was measured at 562 nm. In both assays, ethylenediamine tetraacetic acid (EDTA 1 mg/mL), a synthetic metal chelator, was used as a positive control.

### 3.8. Evaluation of the FA Profile of the Biomass

#### 3.8.1. FAME Preparation

Lipids were converted into the corresponding FAMEs according to the method described by Lepage and Roy [80]. In brief, 1.5 mL of the derivatization solution (methanol/acetyl chloride, 20:1, *v*/*v*) were mixed with 100 mg of dried algae. After 15 min at RT in an ultrasound-water bath, 1 mL of hexane was added to the samples and then heated for one hour at 100 °C. One milliliter of distilled water was added to the transesterification solution after cooling in an ice bath, and the organic phase was removed and dried with anhydrous sodium sulphate. The extracts were then filtered (0.2 µm) and the FAMEs profile determined by GC/MS [81].

#### 3.8.2. Determination of FAMEs Profile by GC–MS

An Agilent GC–MS (Agilent Technologies 6890 Network GC System, 5973 Inert Mass Selective Detector) equipped with a ZB-5MS capillary column (30 m × 0.25 mm internal diameter, 0.25 µm film thickness, Agilent Tech) with helium as the carrier gas was used to determine the FAMEs profile. The injection of samples occurred at 300 °C with a temperature profile of the GC oven at 60 °C (1 min), 30 °C min^−1^ to 120 °C, 4 °C min^−1^ to 250 °C, and 20 °C min^−1^ to 300 °C (4 min). Total ion mode was used for the identification and quantification of FAMEs, and the A—Supelco^®^ 37 Component FAME Mix‖(Sigma-Aldrich, Sintra, Portugal) was used as a standard. Different calibration curves were generated for each of the FAME in this standard and results were expressed as mg/g of dry weight (DW).

### 3.9. Statistical Analysis

All the experiments were conducted at least in triplicate and results were expressed as mean ± standard deviation (SD). Significant differences (*p* < 0.05) were assessed by one-way analysis of variance (ANOVA). If significant, Tukey’s pairwise multiple comparison tests were applied. *R*-statistics^®^ 3.5.3 version was used for the statistical analyses.

## 4. Conclusions

In this work, we explored two strains of microalgae: naviculoid diatom (IMA053) and *T. marina* (IMA043), as sources of natural antioxidants, phenolic compounds, and FAs. Both strains were identified by molecular and morphological analyses before the bioprospecting of these organisms. Overall, the obtained results suggest that DCM extracts from both species have a relevant antioxidant activity against DPPH and ABTS radicals. The naviculoid diatom (IMA053) had a higher iron and copper chelating ability. The extracts of both microalgae were rich in phenolics and a positive correlation between the TCP and the RSA was found, indicating that these algae could be a promising source for nutraceutical and pharmaceutical industries. Moreover, our results suggest that the high proportion of SFAs and MUFAs could render IMA053 and IMA043 desirable feedstocks for biofuel production. The obtained results could be a springboard on these microalgal strains for developing further studies, such as biochemical variation on the phenolics and lipid production and on the FAs profile under different growing conditions. Future steps in accomplishing the study of the strains IMA043 and IMA053 will be the evaluation of the chemical profiles of the extracts by LC/MS/MS and the assessment of their bioactivity through antimicrobial, enzyme inhibitory, and cytotoxicity assays.

## Figures and Tables

**Figure 1 marinedrugs-20-00207-f001:**
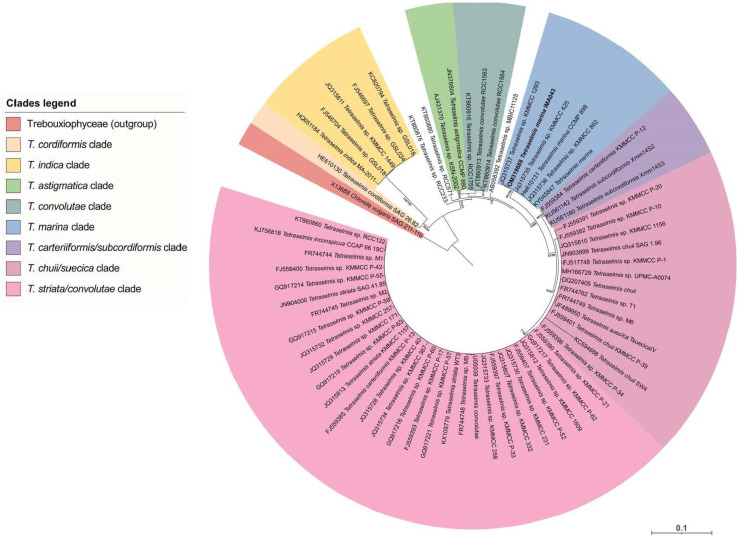
Maximum likelihood (ML) phylogeny based on the 18S rRNA for IMA043-aligned dataset. Approximate Likelihood Ratio Test (SH-aLRT) and ML bootstrap support values (BP) are indicated at nodes. Approximate Likelihood Ratio Test and bootstrap supports ≥70% are shown. Scale number indicates substitutions/site.

**Figure 2 marinedrugs-20-00207-f002:**
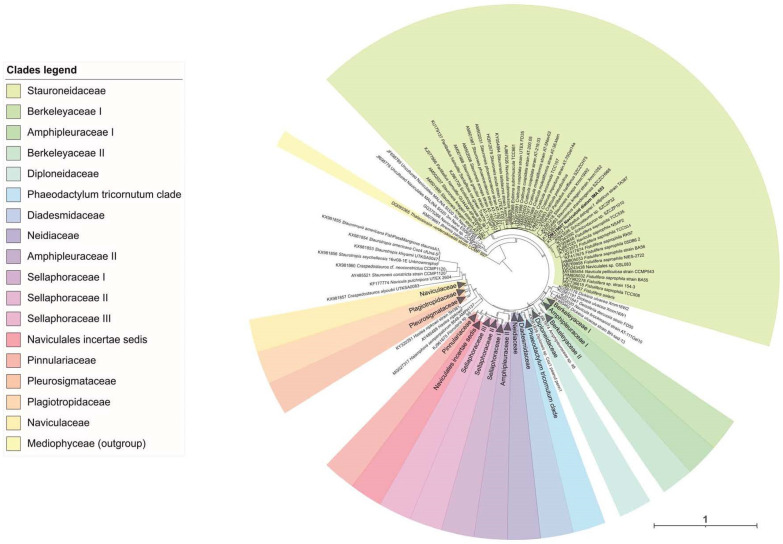
Maximum likelihood (ML) phylogeny based on the 18S rRNA for IMA053 aligned dataset. Approximate Likelihood Ratio Test (SH-aLRT) and ML bootstrap support values (BP) are indicated at nodes. Approximate Likelihood Ratio Test and bootstrap supports ≥70% are shown. Scale number indicates substitutions/site.

**Figure 3 marinedrugs-20-00207-f003:**
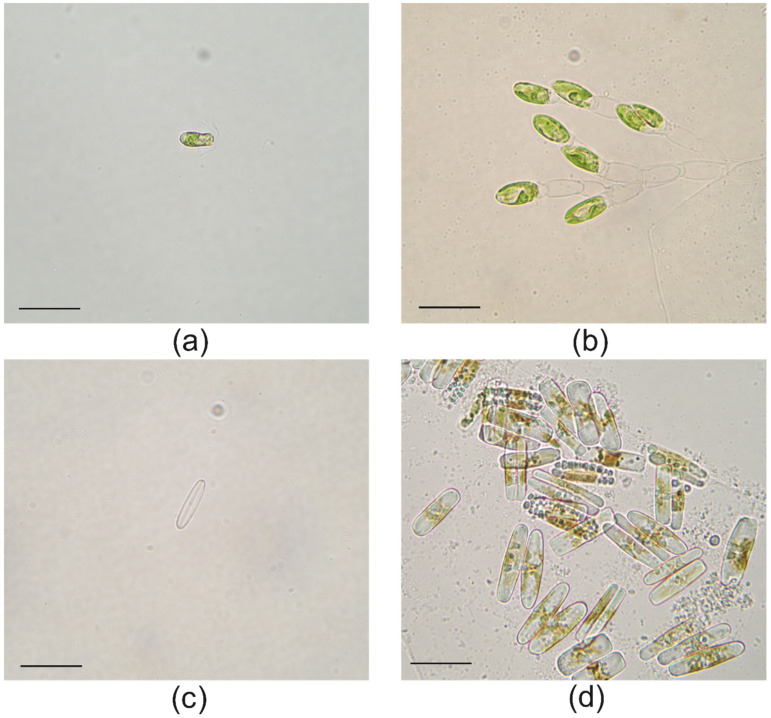
Light microscopy image of the strains IMA043 and IMA053: (**a**) single cell of *T. marina* IMA043, scale bar = 20 µm; (**b**) nonmotile vegetative stage of *T. marina* IMA043, scale bar = 20 µm; (**c**) valve view of a single cell of the naviculoid diatom IMA053, scale bar = 20 µm; (**d**) lipid-rich cells of naviculoid diatoms IMA053, scale bar = 20 µm.

**Figure 4 marinedrugs-20-00207-f004:**
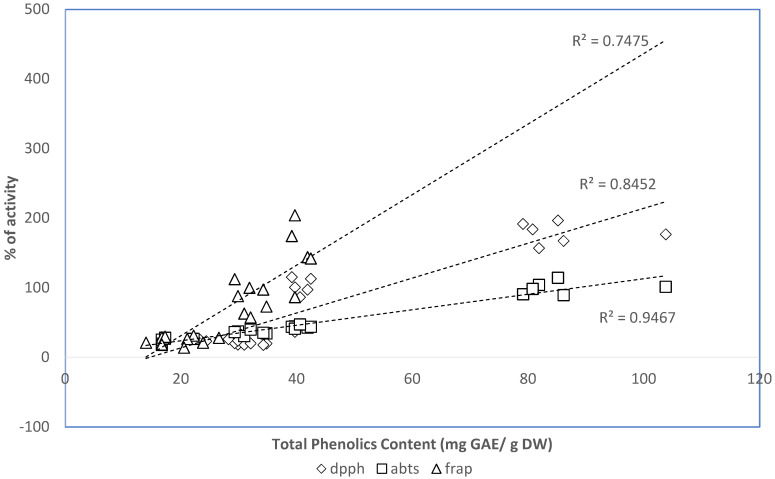
Correlation between the antioxidant capacity and the total phenolic content of the ethanol extracts of naviculoid diatom (IMA053) and *T. marina* (IMA043).

**Table 1 marinedrugs-20-00207-t001:** Total phenolic content (TPC, mg GAE/100 g DW, GAE: gallic acid equivalents) of different extracts of *T. marina* (IMA043) and naviculoid diatom (IMA053).

	Extract	TPC
*T. marina* (IMA043)	Water	20.45 ± 1.87 ^d^
	Methanol	25.19 ± 1.26 ^cd^
	DCM	86.14 ± 4.52 ^a^
Naviculoid diatom (IMA053)	Water	17.80 ± 0.88 ^d^
	Methanol	31.85 ± 0.92 ^c^
	DCM	40.58 ± 0.66 ^b^

Data represent the mean ± standard error of mean (SEM) (*n* ≥ 6). In each column, different letters mean significant differences (Multiple Comparisons of Means: Tukey Contrast, 95% family-wise confidence level).

**Table 2 marinedrugs-20-00207-t002:** Radical scavenging on DPPH and ABTS radicals of different extracts from naviculoid diatom (IMA053) and *T. marina* (IMA043). Results are expressed as antioxidant activity (% activity).

	Extract	ABTS			DPPH		
		1 mg/mL	5 mg/mL	10 mg/mL	1 mg/mL	5 mg/mL	10 mg/mL
Naviculoid diatom (IMA053)	Methanol	n.a.	23.37 ± 0.58 ^c^	35.54 ± 1.28 ^b^	n.a.	12.53 ± 0.44 ^c^	18.91 ± 0.46 ^c^
	Water	n.a.	11.06 ± 1.27 ^d^	23.51 ± 1.81 ^c^	n.a.	n.a.	n.a.
	DCM	10.09 ± 0.53 ^b^	46.91 ± 0.42 ^b^	43.83 ± 0.84 ^b^	18.25 ± 6.08 ^b^	41.55 ± 2.47 ^b^	91.38 ± 11.80 ^b^
*T. marina* (IMA043)	Methanol	n.a.	n.a.	n.a.	n.a.	17.29 ± 0.35 ^c^	24.94 ± 0.71 ^c^
	Water	n.a.	n.a.	n.a.	n.a.	n.a.	n.a.
	DCM	24.66 ± 0.32 ^a^	88.19 ± 2.41 ^a^	99.65 ± 3.76 ^a^	40.96 ± 0.94 ^a^	103.43 ± 7.12 ^a^	178.75 ± 6.13 ^a^
	BHT *	94.77 ± 0.04			80.07 ± 0.72		

Values represent the mean ± standard error of mean (SEM) (*n* ≥ 6). n.a: no activity; * positive control tested at 1 mg/mL; DCM: dichloromethane. For the same column, different letters are significantly different (Multiple Comparisons of Means: Tukey Contrast, 95% family-wise confidence level).

**Table 3 marinedrugs-20-00207-t003:** Ferric reducing antioxidant power (FRAP) of different extracts from naviculoid diatom (IMA053) and *T. marina* (IMA043). Results are expressed as antioxidant activity (% activity).

	Extract	FRAP		
		1 mg/mL	5 mg/mL	10 mg/mL
Naviculoid diatom (IMA053)	Methanol	22.01 ± 0.68 ^c^	60.68 ± 3.54 ^ab^	70.23 ± 5.47 ^b^
	Water	n.a.	23.29 ± 4.24 ^b^	26.75 ± 1.73 ^c^
	DCM	46.35 ± 8.26 ^ab^	135.10 ± 45.63 ^a^	150.10 ± 17.85 ^a^
*T. marina* (IMA043)	Methanol	36.17 ± 2.04 ^bc^	92.28 ± 4.13 ^ab^	103.17 ± 3.26 ^b^
	Water	n.a.	19.56 ± 2.15 ^b^	21.47 ± 2.14 ^c^
	DCM	67.16 ± 3.46 ^a^	40.02 ± 5.62 ^ab^	n.a.

Values represent the mean ± standard error of mean (SEM) (*n* ≥ 6). n.a: no activity; DCM: dichloromethane. For the same column, different letters are significantly different (Multiple Comparisons of Means: Tukey Contrast, 95% family-wise confidence level).

**Table 4 marinedrugs-20-00207-t004:** Metal chelating activities on copper (CCA) and iron (ICA) of different extracts from naviculoid diatom (IMA053) and *T. marina* (IMA043).

	Extract	ICA			CCA		
		1 mg/mL	5 mg/mL	10 mg/mL	1 mg/mL	5 mg/mL	10 mg/mL
Naviculoid diatom (IMA053)	Methanol	n.a.	n.a.	n.a.	n.a.	n.a.	n.a.
	Water	n.a.	83.43 ± 2.22 ^a^	92.05 ± 3.54 ^a^	13.92 ± 2.39 ^b^	29.23 ± 5.53 ^a^	35.31 ± 3.89 ^b^
	DCM	n.a.	49.18 ± 3.77 ^b^	63.57 ± 5.34 ^b^	n.a.	21.11 ± 8.01 ^a^	67.48 ± 14.68^a^
*T. marina* (IMA043)	Methanol	34.12 ± 3.00	n.a.	n.a.	n.a.	n.a.	n.a.
	Water	n.a.	59.25 ± 4.23 ^b^	68.00 ± 6.13 ^b^	21.00 ± 2.88 ^a^	25.92 ± 3.41 ^a^	28.87 ± 1.57 ^b^
	DCM	n.a.	n.a.	n.a.	n.a.	n.a.	11.58 ± 1.49 ^c^
	EDTA *	72.28 ± 2.79			89.69 ± 0.55		

Values represent the mean ± standard error of mean (SEM) (*n* ≥ 6). n.a: no activity; * positive control tested at 1 mg/mL; DCM: dichloromethane. For the same column, different letters are significantly different (Multiple Comparisons of Means: Tukey Contrast, 95% family-wise confidence level).

**Table 5 marinedrugs-20-00207-t005:** Fatty acid profile of the microalgae *T. marina* (IMA043) and the naviculoid diatom (IMA053). Values are given as means of total FAME percentage ± standard deviation (*n* = 3). n.d., not detected.

Fatty Acid	Common Name	*T. marina* (IMA043)	Naviculoid Diatom (IMA053)
**∑SFA**		**38.28 ± 2.00**	**45.70 ± 1.28**
(C14:0)	Methyl myristate	0.03 ± 0.08	3.15 ± 0.08
(C15:0)	Methyl pentadecanoate	n.d.	1.50 ± 0.96
(C16:0)	Methyl palmitate	37.74 ± 1.99	40.68 ± 0.83
(C18:0)	Methyl stearate	0.42 ± 0.12	0.37 ± 0.11
(C24:0)	Methyl lignocerate	0.09 ± 0.09	n.d.
**∑MUFA**		**45.41 ± 1.75**	**49.81 ± 1.00**
(C16:1)	Methyl palmitoleate	0.71 ± 0.20	47.25 ± 0.92
(C18:1*n*9c)	*cis*-9-Oleic acid methyl ester	37.52 ± 0.49	0.51 ± 0.15
(C18:1*n*9t)	*trans*-9-Elaidic acid methyl ester	3.17 ± 1.23	1.03 ± 0.29
(C20:1)	Methyl cis-11-eicosenoate	4.01 ± 1.12	n.d.
(C22:1*n*9)	Methyl erucate	n.d.	0.60 ± 0.18
(C24:1*n*9)	Methyl nervonate	n.d.	0.42 ± 0.13
**∑PUFA**		**16.31 ± 1.64**	**4.57 ± 0.47**
(C19:3*n*3)	Methyl linolenate	2.38 ± 1.43	n.d.
(C18:2*n*6c)	Methyl linoleate	11.01 ± 0.29	0.08 ± 0.08
(C20:4*n*6)	*cis*-5,8,11,14-Eicosatetraenoic acid methyl ester	n.d.	1.06 ± 0.34
(C20:5*n*3)	*cis*-5,8,11,14,17-Eicosapentaenoic acid methyl ester	2.50 ± 0.72	3.43 ± 0.32
(C20:3*n*3)	*cis*-11,14,17-Eicosatrienoic acid methyl ester	0.41 ± 0.22	n.d.
(C20:2)	*cis*-11,14-Eicosadienoic acid methyl ester	0.02 ± 0.03	n.d.
**∑*n*-3**		**5.28 ± 1.61**	**3.43 ± 0.32**
**∑*n*-6**		**11.01 ± 0.29**	**1.14 ± 0.35**
**∑*n*-6/∑*n*-3**		**2.08**	**0.33**
**PUFA/SFA**		**0.43**	**0.10**

## Data Availability

Not applicable.

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
