# Peer review of "Total Phenolic Levels, In Vitro Antioxidant Properties, and Fatty Acid Profile of Two Microalgae, Tetraselmis marina Strain IMA043 and Naviculoid Diatom Strain IMA053, Isolated from the North Adriatic Sea"

_marinedrugs, 2022, doi:10.3390/md20030207_

Round 1

Reviewer 1 Report

This study evaluates the nutritional and biofuel values of two new isolated microalgae, Tetraselmis marina strain IMA043 and naviculoid diatom strain IMA053. This study is meaningful for exploiting new microalgal source in different sectors. However, it still needs to be improved after minor revision.

  1. The manuscript is well written, but there still exits several grammar and format problems, such as “Results suggested” in Abstract, and Parentheses and Spaces in introduction. In addition, sentence as “The DCM extracts…of the same species (92.05% at 10 mg/mL)” in Abstract is suggested to divide into two sentences for clear expression.
  2. Better to provide the biomass concentration is the Results, the concentration is the key index for the exploitation of high-value products from microalgae.
  3. Better to adjust the order of “Fatty acid profile, in vitro antioxidant properties and total phenolic levels” in Title and Abstract, as the fatty acid profile is evaluated at the last step in the text.
  4. Better to discuss more information about the biofuel potential of these two microalgae including the C16:0 and C18:1, which is responsible for lipid stabilization and heat of combustion.

Author Response

Response to Reviewer 1 Comments

Comment #1. The manuscript is well written, but there still exits several grammar and format problems, such as “Results suggested” in Abstract, and Parentheses and Spaces in introduction. In addition, sentence as “The DCM extracts…of the same species (92.05% at 10 mg/mL)” in Abstract is suggested to divide into two sentences for clear expression.

Answer: The manuscript was extensively revised and corrected, according to the reviewer remarks.

Comment #2. Better to provide the biomass concentration is the Results, the concentration is the key index for the exploitation of high-value products from microalgae.

Answer: This type of investigation was not among the aims of this work, which is a preliminary screening of the two strains of the North Adriatic Sea. However, your suggestion is remarkable. We will certainly take this into account for future more in-depth studies on these species, starting from the evaluation of the effects of different environmental factors (such as nutrient variation in the culture medium, temperature, salinity) on the concentration of active compounds.

Comment #3. Better to adjust the order of “Fatty acid profile, in vitro antioxidant properties and total phenolic levels” in Title and Abstract, as the fatty acid profile is evaluated at the last step in the text.

Answer: The order was corrected in the title and in the abstract, as suggested.

Comment #4. Better to discuss more information about the biofuel potential of these two microalgae including the C16:0 and C18:1, which is responsible for lipid stabilization and heat of combustion.

Answer: Following the reviewer’s suggestion, the discussion of the biofuel potential of the two strains of microalgae was expanded.

Reviewer 2 Report

1.    Why those two microalga species were selected? Do they have something in common?
2.    Significancy and novelty of the current research is not completely clear and must be added into Introduction. 
3.    What is the purpose of Figures 1 and 2. If it was clearly stated that the species were identified, there is no need in adding two diagrams with the full taxonomy of species?
4.    Item 2.3 is irrelevant. First paragraph is not about the phenolic contents. Polyphenols is a vast group, which members provide any kind of biological activity. Statement about observed differences in phenolic contents due to different methods used suggest the low accuracy of experiments. It would be good to explain how the results of the present study can be trustworthy because methodology should not affect the final results and conclusion of the study.
5.    Item 2.4 provides huge amounts of information regarding antioxidant activity with more 90% is just a general knowledge, which is not linked to the objects of the study. The results obtained are not discussed. 
6.    Item 2.6 just the same – too much other finding that are not directly linked to objects of present study, and not enough discussions of the current results.

Author Response

Response to Reviewer 2 Comments

Comment #1.  Why those two microalga species were selected? Do they have something in common?  

Answer: The microalgal strains investigated in this study were obtained from seawater samples collected during sampling campaigns aimed at screening the biodiversity of North Adriatic phytoplankton and at surveying the potential of microalgal strains for biotechnological purposes. The selected strains were among those which were succesfully isolated and grown in culture, with the production of enough biomass to carry out the planned experiments. Once in culture, the two strains were also taxonomically identified. Thus, there was not a preliminary species selection, even if the comparison of two microalgal strains from different taxonomic groups (the phylum Chlorophyta, and more precisely the genus Tetraselmis, and the phylum Bacillariophyta, and more precisely order Naviculales, resepctively), already known for their biotechnological potential (as reported in the Introduciton), was surely considered.

Comment #2. Significancy and novelty of the current research is not completely clear and must be added into Introduction. 

Answer: As required, we added a paragraph about this at the end of the Introduction.

Comment #3. What is the purpose of Figures 1 and 2. If it was clearly stated that the species were identified, there is no need in adding two diagrams with the full taxonomy of species? 

Answer: Even if we compared the 18S rRNA sequences obtained from the two investigated strains with the sequences available in Genbank using the BLAST tool, we think that the phylogenetic reconstructions of Figures 1 and 2 are useful to illustrate the taxonomic position of the two investigated microalgae. This is particularly true for the naviculoid diatom. In fact, if for the Chlorophyta its belonging to the genus Tetraselmis was already clear based on morphology and was confirmed by the molecular identity of its 18S rRNA sequence (which allowed also its attribution to the species T. marina), the same could not be said for the naviculoid diatom. Indeed, for the diatom the morphology clearly indicated its belonging to the Naviculales order, but the BLAST search did not allow to give a more precise taxonomic position to the organism, whose sequences shared similar identities with different naviculoid genera, as well as with a not better identified strain. Thus, in the case of the diatom, the phylogenetic reconstruction (which is based not just on the sequence identities, but on an evolutionary model) was required to try to understand if it could be attributed to one of the genera found with the BLAST search and allowed us to clearly state that the diatom belonged to the Stauroneidaceae family. In order to show similar data for both the investigated strains, we think that it is better to keep both the tree figures.

Comment #4. Item 2.3 is irrelevant. First paragraph is not about the phenolic contents. Polyphenols is a vast group, which members provide any kind of biological activity. Statement about observed differences in phenolic contents due to different methods used suggest the low accuracy of experiments. It would be good to explain how the results of the present study can be trustworthy because methodology should not affect the final results and conclusion of the study.

Answer: The first paragraph is used to put in context the importance of quantifying the levels of phenolic compounds, in algae, and was simplified, for clarity. In the text we state that our results are different than those reported in the literature for related species (not the same species), and that those differences could be ascribed to the different solvents used for the extraction (since it is known that different solvents extract different types and levels of polyphenolic compounds), or by the different growing conditions used for the cultivation of the microalgae (it is known that the production and accumulation of polyphenolic compounds is highly dependent on the culture conditions of the microalgae). The paragraph including this information was modified, for clarity.

Comment #5 Item 2.4 provides huge amounts of information regarding antioxidant activity with more 90% is just a general knowledge, which is not linked to the objects of the study. The results obtained are not discussed. 

Answer: The first sentence of section 2.4 was shortened, and the text modified, for clarity. The authors consider that the results are discussed in both section 2.4 and 2.5.

Comment #6.   Item 2.6 just the same – too much other finding that are not directly linked to objects of present study, and not enough discussions of the current results.

Answer: This section was revised and discussion was improved, as suggested.